# PARP-1 is a transcriptional rheostat of metabolic and bivalent genes during development

Gbolahan Bamgbose, Alexei Tulin

**PARP-1 participates in various cellular processes, including gene regulation. In *Drosophila*, PARP-1 mutants undergo developmental arrest during larval-to-pupal transition. In this study, we investigated PARP-1 binding and its transcriptional regulatory role at this stage. Our findings revealed that PARP-1 binds and represses active metabolic genes, including glycolytic genes, whereas activating low-expression developmental genes, including a subset of "bivalent" genes in third-instar larvae. These bivalent promoters, characterized by dual enrichment of low H3K4me3 and high H3K27me3, a unimodal H3K4me1 enrichment at the transcription start site (conserved in *C. elegans* and zebrafish), H2Av depletion, and high accessibility, may persist throughout development. In PARP-1 mutant third-instar larvae, metabolic genes typically down-regulated during the larval-to-pupal transition in response to reduced energy needs were repressed by PARP-1. Simultaneously, developmental and bivalent genes typically active at this stage were activated by PARP-1. In addition, glucose and ATP levels were significantly reduced in PARP-1 mutants, suggesting an imbalance in metabolic regulation. We propose that PARP-1 is essential for maintaining the delicate balance between metabolic and developmental gene expression programs to ensure proper developmental progression.**

## Introduction

Gene regulation is a fundamental process that controls the normal development and function of living organisms. During development, a complex series of molecular interactions guide the transformation of a single cell into a multicellular organism with a diverse array of cell types and structures. The orchestration of these events is primarily driven by the precise regulation of gene expression, ensuring that the right genes are activated or silenced at the appropriate times and in the correct tissues across a wide range of species, from single-celled organisms to plants and animals.

The regulation of gene expression is orchestrated through the synergistic action of a plethora of regulators, including signaling molecules, transcription factors, and chromatin remodelers. Transcription factors modulate gene expression by binding to specific DNA sequences (Spitz & Furlong, 2012), and signaling molecules typically influence gene expression indirectly through the activation of signaling pathways that ultimately impact transcription factors (Hunter, 2000; Kabir et al, 2018). Chromatin remodelers, on the other hand, ensure precise control over the activation or repression of target genes by modifying the chromatin structure (Ho & Crabtree, 2010).

One such factor is PARP-1, a transcriptional regulator and chromatin-associated poly(ADP-ribose) polymerase that is crucial for normal development (Tulin et al, 2002; Ménissier de Murcia et al, 2003). In mice, PARP-1 knockouts are viable. However, double knockout of both PARP-1 and PARP-2 causes embryonic lethality, suggesting functional redundancy between the two PARPs (Ménissier de Murcia et al, 2003). In another study, conditional knockout or catalytic inhibition of PARP-1 and PARP-2 led to decidualization failure and pregnancy loss in mice (Kelleher et al, 2021). In *Drosophila*, the importance of PARP-1 in larval development and the transition to adulthood has been demonstrated through various studies on mutants and RNAi-mediated knockdowns. For instance, depletion of maternal stores of *Parp* RNA via RNAi in early-stage embryos results in developmental arrest at the first-instar larvae stage (Tulin et al, 2002). *Parp*[CH1] mutants, generated via insertional *P* element mutagenesis, die at the second instar stage (Zhang & Spradling, 1994; Tulin et al, 2002). In addition, *Parp*[C03256] hypomorph mutants express a significantly reduced and truncated PARP-1 protein lacking its first zinc finger (Kotova et al, 2010). These *Parp*[C03256] mutants are arrested in development during larval-to-pupal transition, with only rare escapers surviving to the late pharate stage (Kotova et al, 2010). Collectively, these findings underscore the essential role of PARP-1 in the development of various organisms, including both *Drosophila* and mice, and emphasize its importance in the transition to adulthood or other developmental stages.

We previously demonstrated that PARP-1 remodels chromatin via poly(ADP-ribosyl)ation during normal development (Tulin & Spradling, 2003) and maximal accumulation of ADP-ribose polymers is highest at

---

Department of Biomedical Sciences, School of Medicine and Health Sciences, University of North Dakota, Grand Forks, ND, USA

Correspondence: alexei.tulin@und.edu

 

the late third-instar and prepupal stages (Kotova et al, 2009), suggesting that PARP-1 may regulate developmentally regulated genes needed to transition from larval stages to adulthood. Therefore, we used *Drosophila*, the genome of which encodes a single *Parp* gene, to uncover the transcriptional role of PARP-1 during development.

In this study, we investigated the role of PARP-1 in regulating gene expression during development in *Drosophila*. We examined PARP-1 binding and its impact on transcriptional regulation, demonstrating that PARP-1 suppresses active metabolic genes, which are typically down-regulated during larval-to-pupal transition, whereas activating less active developmental and bivalent genes that are expected to be up-regulated. Our findings reveal that PARP-1 is essential for maintaining the balance between metabolic and developmental gene expression, enabling proper progression through developmental stages. Furthermore, we provide evidence of disrupted metabolic regulation in PARP-1 mutants. These results underscore the importance of PARP-1 as a key regulator of development.

## Results

### PARP-1 binds the promoters of highly expressed genes

We performed chromatin immunoprecipitation to determine PARP-1-binding sites in wandering *Drosophila* third-instar larvae (L3) using a YFP-tagged PARP-1 construct. We identified 12,963 PARP-1-binding peaks (FDR < 0.05) with 55%, 33%, and 9% of these peaks in promoters (±500 bp transcription start site [TSS]), gene bodies, and distal intergenic regions, respectively (Figs 1A and S1A and B and Table S1).

To understand PARP1's association with chromatin features, we analyzed public *Drosophila* datasets (Table S2). PARP-1 peaks were strongly correlated and colocalized with active chromatin signatures (Pol II, H2Av, H3K4me3, H3K9ac, H3K27ac, and ATAC-seq), but less associated with repressive histone marks (H3K27me3, H3K9me2, and H3K9me3) in third-instar larvae (Fig 1B). We assigned PARP-1 peaks to annotated genes or the gene closest to the peak. PARP-1 peaks were more associated with highly expressed genes compared with low-expression and silent gene groups (Fig 1C). Furthermore, metagene plots of normalized PARP-1 ChIP-seq signals showed that PARP-1 mostly occupied the promoters of high-expression genes and highly accessible genes (expression quartiles: Q1 and Q2) (Fig 1E). Taken together, our results show that PARP-1 predominantly binds transcriptionally permissive promoters in *Drosophila* third-instar larvae, which is consistent with previous studies in human and mouse cells (Kraus, 2008; Krishnakumar & Kraus, 2010; Liu & Kraus, 2017).

### PARP-1 regulates divergent gene expression programs by repressing highly active metabolic genes and activating developmental genes

With PARP-1 occupancy at high-expression genes, we asked if it activates or represses their expression. To address this question, we performed RNA-seq in *Parp^C03256* hypomorph mutants to integrate

PARP-1-binding data with its transcriptional activity (Table S3). We examined PARP-1 occupancy and enrichment of chromatin signatures in WT animals at active genes (Q1+Q2; Fig 1D and E) differentially expressed in *Parp^C03256* third-instar larvae. PARP-1 occupancy at active genes, both down- and up-regulated in *Parp^C03256*, was comparable in WT animals (Fig 2A). Active chromatin signatures were highly enriched at up-regulated genes in *Parp^C03256* compared with down-regulated genes (Fig 2A). However, in WT animals, ATAC-seq signals were mostly the same at active genes differentially expressed in *Parp^C03256* (Fig 2A). In contrast to up-regulated genes, down-regulated genes in *Parp^C03256* were more enriched with repressive histone marks, particularly H3K27me3, compared with up-regulated genes (Fig 2A). However, PARP-1 was more associated with H3K27me3 than H3K9me2/3 (Fig 2A). In addition, in WT animals, highly expressed genes up-regulated in *Parp^C03256* animals were, on average, more expressed than down-regulated genes (Fig 2B). Thus, even though down-regulated genes in *Parp^C03256* had a higher enrichment of repressive marks and were less expressed than the genes up-regulated in *Parp^C03256* in WT, they remained highly accessible, a characteristic of bivalent promoters (Ramirez-Carrozzi et al, 2009; Deaton & Bird, 2011; Mas et al, 2018), which are simultaneously marked by opposing histone modifications: H3K4me3 and H3K27me3 (Bernstein et al, 2006). Gene ontology analysis showed that PARP-1 repressed metabolic genes and activated transcription factors required for development, specifically neurogenesis and morphogenesis (Fig 2C). Fig 2D shows examples of genes regulated by PARP-1. The genes encoding Cytochrome c proximal (Cyt-c-p), an electron-carrier protein, and spalt-related salr, a zinc finger transcription factor, were up- and down-regulated in *Parp^C03256*, respectively (Fig 2D). Notably, the salr gene, down-regulated in *Parp^C03256*, has a bivalent profile (Fig 2D; right).

Because PARP-1 represses metabolic genes while activating developmental genes, we asked if high-expression genes repressed by PARP-1 are primarily housekeeping genes, which tend to be shorter in length, whereas genes activated by PARP-1 are mainly developmental control genes or bivalent genes. These genes are typically involved in neurogenesis and morphogenesis, have more introns and regulatory regions, and, therefore, tend to be longer (Stark et al, 2007; Zeitlinger et al, 2007; Zeitlinger & Stark, 2010; Blanco et al, 2020). Indeed, genes up-regulated in *Parp^C03256* were generally shorter than down-regulated genes and a randomly selected group of unchanged genes, whereas down-regulated genes in *Parp^C03256* were longer than up-regulated genes and unchanged genes (Fig S2).

Our findings suggest that PARP-1 binding is involved in bidirectional gene regulation, that is, both up- and down-regulation at actively expressed genes. During the third-instar larvae stage, PARP-1 tempers the expression of highly active metabolic genes whereas, at the same time, it facilitates the derepression and activation of developmental genes through its occupancy.

### A unimodal H3K4me1 enrichment marks bivalent promoters across species

Developmental genes that encode transcription factors and components of signaling pathways are typically enriched with bivalent histone marks on their promoters, catalyzed by the Trithorax

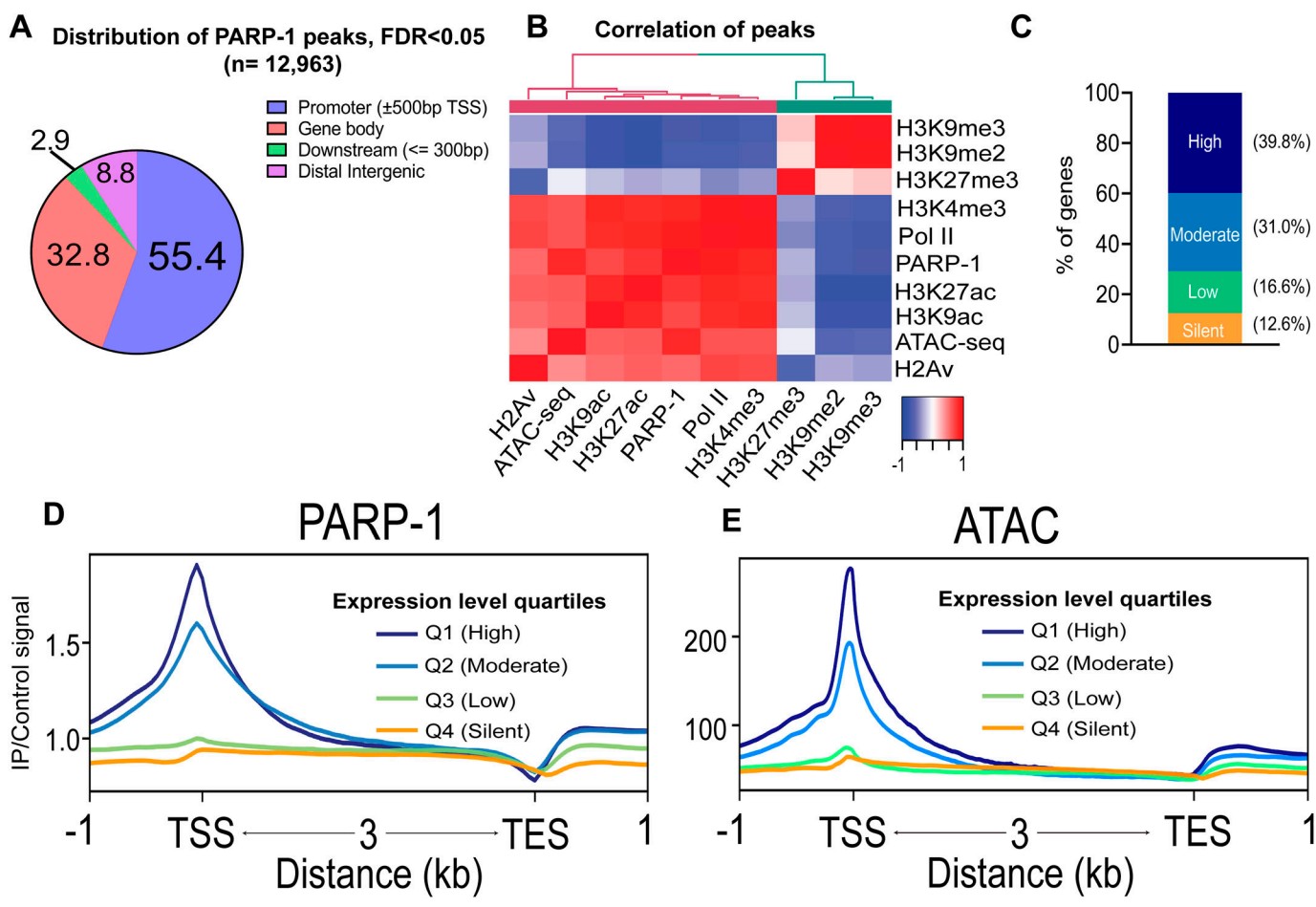

**Figure 1. PARP-1 predominantly occupies the promoters of highly expressed genes.**
**(A)** Pie chart showing the percentage of PARP-1 ChIP-seq peaks in *Drosophila* third-instar larvae across genomic features. **(B)** Heatmap showing spearman correlation of peaks from ChIP-seq overlaps in third-instar larvae. **(C)** Percentage distribution of gene expression levels for genes overlapping PARP-1 peaks and the nearest genes closest to PARP-1 peaks. Genes were categorized by expression levels based on steady-state mRNA expression quartiles (High = 75–100%, Moderate = 50–75%, Low = 25–50%, Silent = 0–25%) in WT third-instar larvae, as determined by RNA-seq analysis. **(D, E)** Metagene plots of normalized PARP-1 ChIP-seq signals and ATAC-seq signals at *Drosophila* genes stratified by steady-state mRNA expression quartiles in WT third-instar larvae. The graph shows PARP-1 and ATAC-seq signals in third-instar larvae at the regions extending from −1 kb of the transcription start site to +1 kb of the transcription end site.

group/MLL2 (H3K4me3), and Polycomb group (H3K27me3), specifically, the Polycomb repressive complex 2. Gene bivalency was first described in mouse embryonic stem cells (ES), where bivalent genes are thought to be poised for future activation or silencing, depending on developmental cues (Azuara et al, 2006; Bernstein et al, 2006). Bivalent promoters were also found in zebrafish (Vastenhouw et al, 2010), humans (Pan et al, 2007; Zhao et al, 2007), and, debatably, in *Drosophila* (Schertel et al, 2015; Kang et al, 2017; Akmammedov et al, 2019). Despite this model's prevalence, evidence shows that depleting MLL2 complex subunits in ESCs only cause a minimal reduction in gene expression levels (Piunti & Shilatifard, 2016). Furthermore, during retinoic acid differentiation, cells lacking MLL2 can still initiate developmental gene expression programs (Hu et al, 2013a; Denissov et al, 2014). Thus, the purpose of bivalency is still unclear. Our results suggest that PARP-1 may activate bivalent genes during development in third-instar larvae. Thus, we hypothesized that PARP-1 might be required to activate bivalent genes crucial for larval-to-pupal transition.

We annotated promoters in third-instar larvae using peaks of H3K4me3 and H3K27me3. We identified 5281 H3K4me3-only promoters (active), 257 bivalent promoters, and 696 H3K27me3-only promoters (silent) (Table S4). PARP-1 occupancy was high in active and bivalent promoters but depleted in silent promoters (Fig 3A). As expected, these bivalent genes had promoter accessibility comparable with that of H3K4me3-only genes. H3K27me3 was more enriched at bivalent promoters than H3K27me3-only promoters, suggesting that H3K27me3 preferentially counteracts H3K4me3 at bivalent promoters (Fig 3A). Surprisingly, H2Av was not enriched at bivalent genes (Fig 3A), unlike its mammalian ortholog H2A.Z (Creyghton et al, 2008; Goldberg et al, 2010; Ku et al, 2012; Hu et al, 2013b).

We detected a unimodal H3K4me1 at the TSS of bivalent genes, a bimodal pattern at active genes, and depletion of H3K4me1 at silent genes (Fig 3A). When examining bivalent genes in whole organisms, it is crucial to account for the potential influence of fluctuating levels of activating (H3K4me3) and repressive (H3K27me3) histone

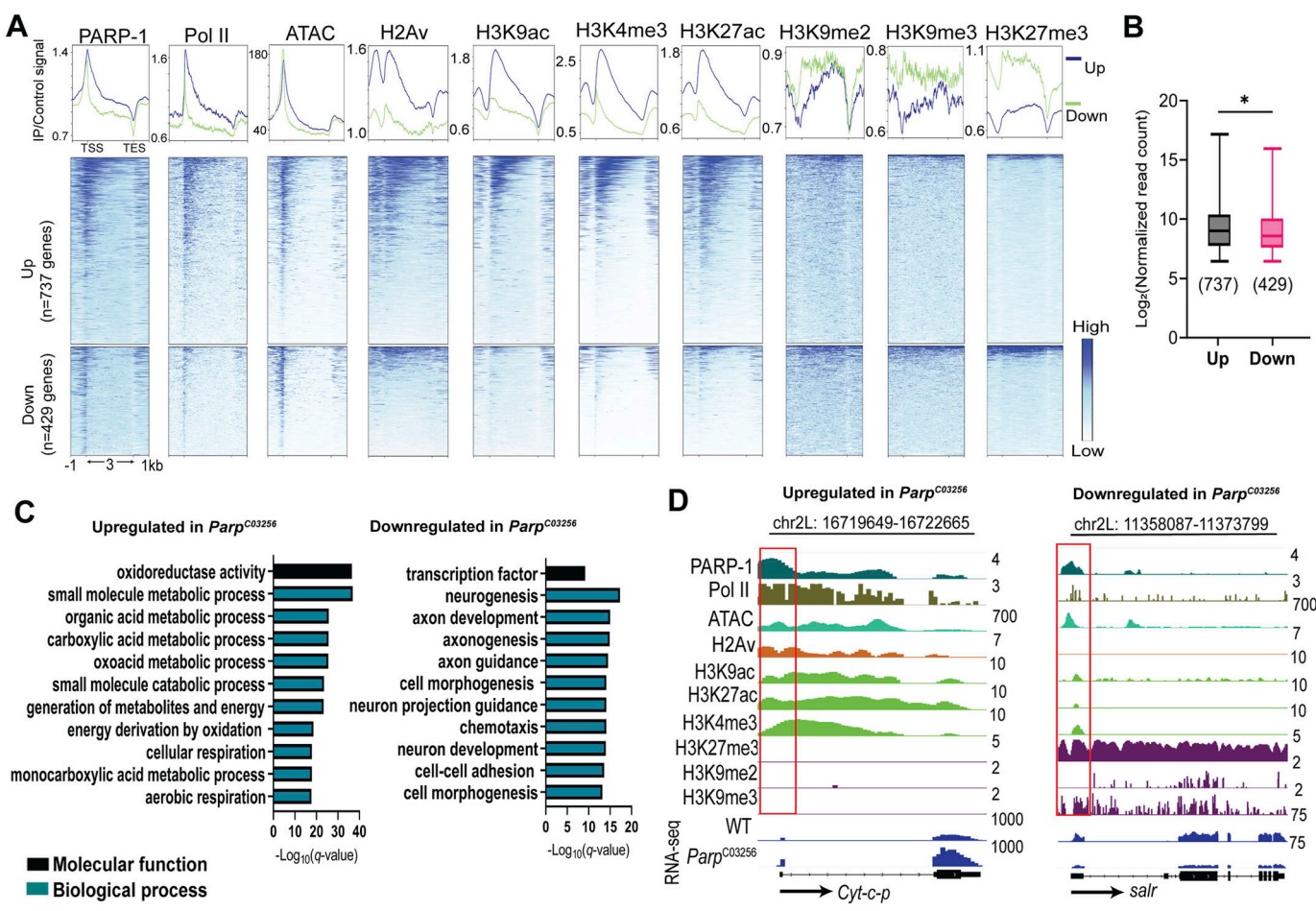

**Figure 2. Highly enriched PARP-1 represses metabolic genes and activates developmental genes.**
**(A)** Heatmaps showing enrichment of normalized PARP-1, Pol II, H2Av, H3K9ac, H3K4me3, H3K27ac, H3K9me2, H3K9me3, H3K27me3 ChIP-seq signals, and ATAC-seq signals in WT third-instar larvae at active genes that were differentially expressed in $Parp^{C03256}$ (up-regulated = 737, down-regulated = 429) third instar larvae and highly occupied by PARP-1. Upper plots show the summary of signals (metagene plot). The graph shows ChIP-seq and ATAC-seq signals in third-instar larvae at the regions extending from −1 kb of the TSS to +1 kb of the TES. **(B)** Expression levels in WT third-instar larvae of active differentially expressed genes in $Parp^{C03256}$ third-instar larvae that are highly occupied by PARP-1. *$P < 0.05$ (Mann–Whitney test; two-tailed). **(C)** Gene ontology of genes that were highly occupied by PARP-1 and differentially expressed in $Parp^{C03256}$ third-instar larvae. **(D)** IGV (Integrative Genomics Viewer) tracks of normalized PARP-1, Pol II, H2Av, H3K9ac, H3K27ac, H3K4me3, H3K27me3, H3K9me2, H3K9me3 ChIP-seq, WT ATAC-Seq signals, and RNA-seq signals of WT and Parp^{C03256} third-instar larvae in the indicated genes, *Cyt-c-p* and *salr*, which were highly occupied by PARP-1, and up-regulated and down-regulated in $Parp^{C03256}$ third-instar larvae, respectively. Black arrow indicates the direction of transcription. Red boxes highlight promoters.

modifications in diverse cell types such as differentiated and progenitor cells. This complexity can result in ambiguous outcomes when assessing the ChIP-seq signal for a particular gene, as it signifies a composite of all chromatin configurations present throughout the various cell types. Nonetheless, this pattern of H3K4me1 enrichment was consistent with the unimodal trend observed at poised promoters in human and mouse cells where H3K4me1 coincides with both H3K4me3 and H3K27me3 (bivalent), as opposed to the bimodal trend seen at active promoters (H3K4me3-only) where it flanks H3K4me3 (Bae & Lesch, 2020). In some instances, the unimodal H3K4me1 pattern observed at bivalent promoters exhibited a non-punctate distribution but still had a high enrichment of H3K4me1, which was proximal to the TSS than at active promoters (Bae & Lesch, 2020; Yu et al, 2023), consistent with our results in third-instar larvae (Fig 3A). This distinct H3K4me1 pattern was closely associated with promoter CpG islands of

bivalent genes, implying that most bivalent promoters are actually "trivalent" and, in this case, characterized by the simultaneous presence of H3K4me1, H3K4me3, and H3K27me3 (Yu et al, 2023). This singular H3K4me1 enrichment pattern at bivalent genes has been observed in various cell types, including embryonic stem cells, germ cells, and differentiated cells in both humans and mice. (Bae & Lesch, 2020; Yu et al, 2023). Furthermore, we examined ChIP-seq data from other model organisms, specifically zebrafish (Murphy et al, 2018) and *Caenorhabditis elegans* (Jänes et al, 2018) (Tables S5 and S6). Notably, we observed a bimodal H3K4me1 enrichment at active promoters and high unimodal enrichment at bivalent promoters in zebrafish sperm. We also detected a unimodal H3K4me1 enrichment at silent promoters, albeit at lower levels, compared with bivalent promoters (Fig S3) in zebrafish sperm. Similarly, in *C. elegans* embryos, we observed unimodal H3K4me1 patterns marking bivalent promoters and bimodal H3K4me1 patterns

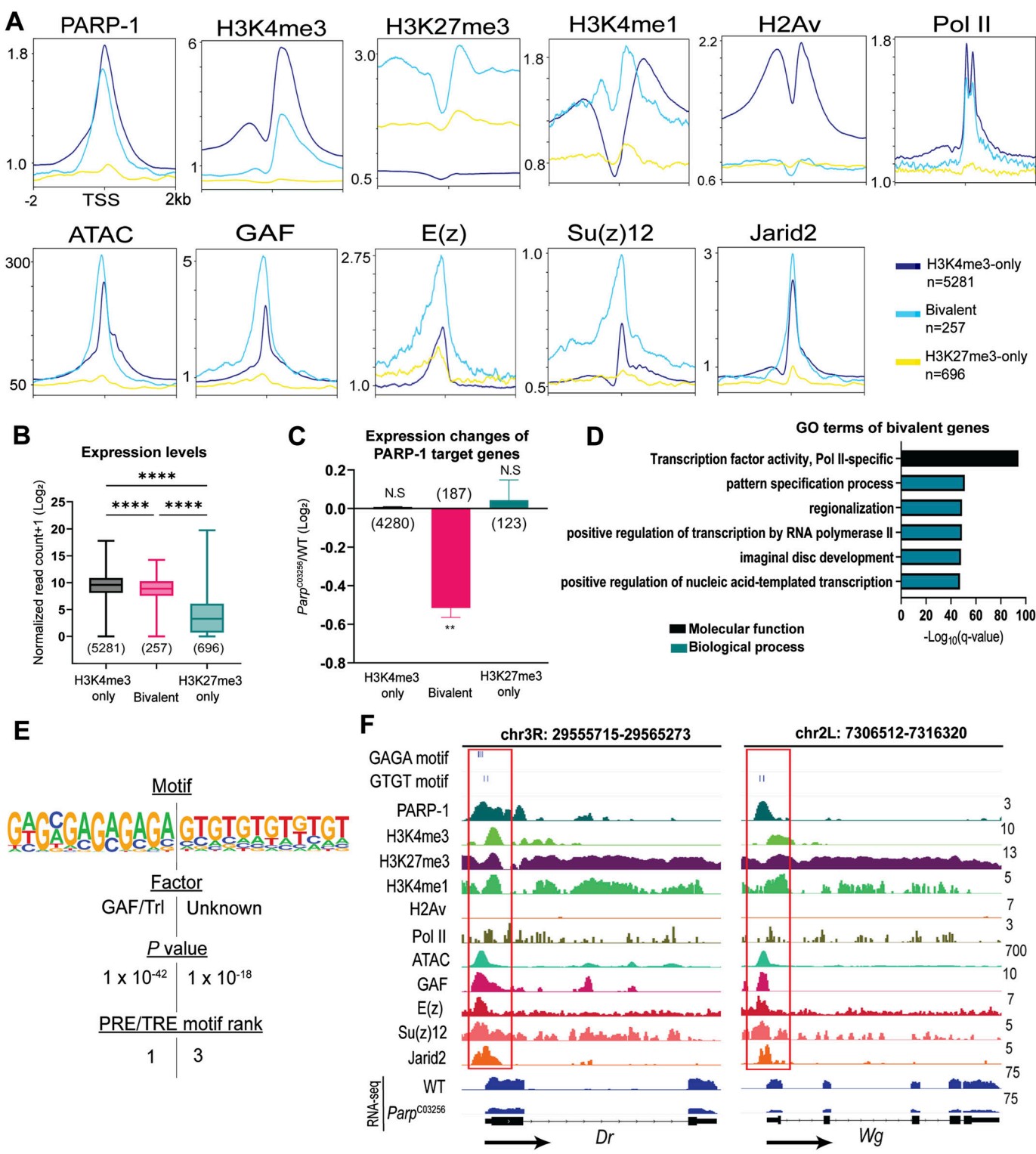

**Figure 3. PARP-1 activates occupied bivalent genes.**
**(A)** Metagene plots showing enrichment of normalized PARP-1, H3K4me3, H3K27me3, H3K4me1, H2Av, Pol II, GAF, E(z), Su(z)12, Jarid2 ChIP-seq signals, and ATAC-seq signals in third-instar larvae at active (H3K4me3-only), bivalent and silent (H3K27me3-only) promoters. ChIP-seq and ATAC-seq signals in third-instar larvae are shown for regions ±2 kb from the TSS. **(B)** Expression levels of active, bivalent, and silent genes in WT third-instar larvae. ****$P < 0.0001$ (Kruskal–Wallis test). Box plot: dashed center line, median; box-plot limits, upper and lower quartiles; whiskers, minimum and maximum values. **(C)** Expression changes of PARP-1-targeted active, bivalent, and silent genes ($Parp^{C03256}$ versus WT) in third-instar larvae. **$P < 0.01$. **(D)** Gene ontology of 257 bivalent genes showing their top molecular function and enriched biological processes. **(E)** HOMER analysis of PARP-1 binding motifs at bivalent promoters. PRE/TRE motif ranks were culled from Ringrose et al (2003). **(F)** IGV tracks of PARP-1,

marking active promoters, whereas H3K4me1 was depleted in silent genes as in *Drosophila* (Fig S3). The zebrafish ortholog of H2A.Z, H2AFV, was highly enriched at bivalent and active promoters in zebrafish sperm, which is in line with previous research findings (Murphy et al, 2018) (Fig S3). In contrast, in *C. elegans* embryos, HTZ-1 was highly enriched at active genes but mostly depleted at bivalent and silent promoters, similar to *Drosophila* (Fig S3). Bivalent promoters are usually enriched at unmethylated CpG islands. Like *Drosophila*, *C. elegans* lacks 5-methylcytosine. Also, they do not have homologs of the enzymes that catalyze DNA 5-cytosine methylation, specifically Dnmt1 and Dnmt3 (Raddatz et al, 2013; Greer et al, 2015). Consequently, depletion of H2A variant at bivalent promoters may be a characteristic specific to *Drosophila*, *C. elegans* and other animals devoid of 5-methylcytosine. Although our findings may not conclusively establish true bivalency, they do suggest that unimodal H3K4me1 patterns and bimodal H3K4me1 patterns are conserved across various species, including *C. elegans*, *Drosophila*, zebrafish, mice, and humans, at poised and active promoters, respectively. Moreover, H3K4me1 might have unique regulatory functions at bivalent genes compared with active ones.

The trxG member, GAF, and three subunits of the polycomb repressive complex 2 , enhancer of zeste E(z), suppressor of zeste 12 Su(z)12 and jumonji AT-rich interactive domain 2, exhibited a higher affinity for the TSS of bivalent promoters, whereas they had a lower occupancy at genes marked only with H3K27me3 (Fig 3A). Genes with bivalent promoters exhibited an intermediate expression profile when compared with active and silent promoters (Fig 3B). Specifically, the expression level of bivalent genes was lower than that of H3K4me3-only genes but higher than that of H3K27me3-only genes (Fig 3B). These observations suggest that genes with bivalent promoters exist in a "poised" state of gene expression.

### PARP-1 binds and activates bivalent genes

We next examined the impact of PARP-1 binding on these genes. PARP-1 occupied the promoters of 4,280 H3K4me3-only genes (81%), 188 bivalent genes (72%), and 123 (~18%) H3K27me3-only genes. Notably, PARP-1-targeted bivalent genes were down-regulated in *Parp^{C03256}* animals, suggesting that PARP-1 directly activates their expression (Fig 3C). GO analysis showed that these bivalent genes were largely transcription factors involved in the "pattern specification process" (Fig 3D).

De novo motif analysis of the promoters of active, bivalent, and silent genes revealed that bivalent promoters were specifically enriched with polycomb/trithorax response element (PRE/TRE) motifs associated with GAF (Strutt et al, 1997), zeste (z) (Saurin et al, 2001), and Adh transcription factor 1 (Orsi et al, 2014) (Fig S4). At bivalent promoters, PARP-1-binding sites were enriched with PRE/TRE motifs (Fig 3E and F), specifically the GAF motif and a "GTGT" motif (Ringrose et al, 2003). HOMER could not identify a transcription factor associated with the "GTGT motif" in *Drosophila*. However, recent findings have revealed that the "GTGT" motif is highly enriched throughout the genome in regions containing

polycomb repressive complex 1 (PRC1) components and polycomb group recruiters (Schuettengruber et al, 2009). Moreover, combgap (cg) has been identified as a binding factor for the "GTGT" motif at PREs and a recruiter of polycomb complexes to a specific subset of PREs (Ray et al, 2016). Thus, our data show that PARP-1 may bind PRE/TREs to activate bivalent genes.

### Bivalency is maintained throughout *Drosophila* development

After annotating bivalent genes in third-instar larvae of *Drosophila*, we proceeded to investigate their enrichment at other developmental stages using public ChIP-seq data (ENCODE). To our astonishment, we discovered the presence of bivalency from the embryonic stage to adulthood, specifically in the heads of mixed adults (Fig 4A). Throughout all developmental stages, bivalent promoters exhibited a high and unimodal enrichment of H3K4me1 at their TSS (Fig 4A). However, the pattern of H3K4me1 enrichment was similar in embryos, third-instar larvae, and adult heads compared with the other developmental stages, which had a more punctate H3K4me1 enrichment at their TSS (Figs 3A and 4A). Again, H2Av was depleted at bivalent promoters in embryos and adults. In addition, the bivalent genes we identified in third-instar larvae also had a "poised" expression profile across various developmental stages in *Drosophila* (Fig 4B). Consequently, our findings suggest that bivalency might persist throughout development in *Drosophila*.

### PARP-1 is not required for the activation of bivalent genes in S2 cells

Next, we characterized the genes identified in third-instar larvae using public CUT&Tag and ChIP-seq data in *Drosophila* S2 cells and Kc167 cells. It is important to note that S2 and Kc167 cells were derived from embryos and are non-clonal (Echalier & Ohanessian, 1969; Schneider, 1972). As a result, the observed genes may not be truly bivalent in the strict sense, and our findings may be influenced by cell heterogeneity.

Despite this limitation, in S2 and Kc167 cells, bivalent genes had a profile similar to that observed at all *Drosophila* stages, including low H3K4me3, high H3K27me3, and a high unimodal H3K4me1 enrichment compared with the bimodal H3K4me1 enrichment at active genes as observed in embryos, third-instar larvae, and adult heads (Fig 5A–C). Also, H2Av was depleted at bivalent promoters in S2 cells (Fig 5D).

Trr, a trithorax protein, and histone H3 lysine 4 monomethylase highly occupied bivalent promoters in S2 cells (Fig 5E). Similarly, TrxG proteins, such as Trithorax-like (GAF/Trl), zeste, trithorax (trx), brahma (brm), absent small or homeotic discs 1 (ash1), and female sterile (1) homeotic (fsh), along with PRC1 proteins, polycomb (Pc), polyhomeotic (Ph), sex combs extra (dRING), and enhancer of zeste (E[z]), exhibited high occupancy at bivalent promoters in S2 cells (Fig 5F–H). Intriguingly, E[z] also occupied active promoters, and (Fig 5H), implying potential roles in active transcription beyond its

H3K4me3, H3K4me1, H2Av, Pol II, GAF, E(z), Su(z)12, Jarid2 ChIP-seq, ATAC-Seq signals in third-instar larvae, and RNA-seq signals of WT and *Parp^{C03256}* third-instar larvae in the indicated bivalent genes, *Dr* and *Wg*, showing enrichment of PRE/TRE motifs. Black arrows indicate the direction of transcription. Red boxes highlight promoters.

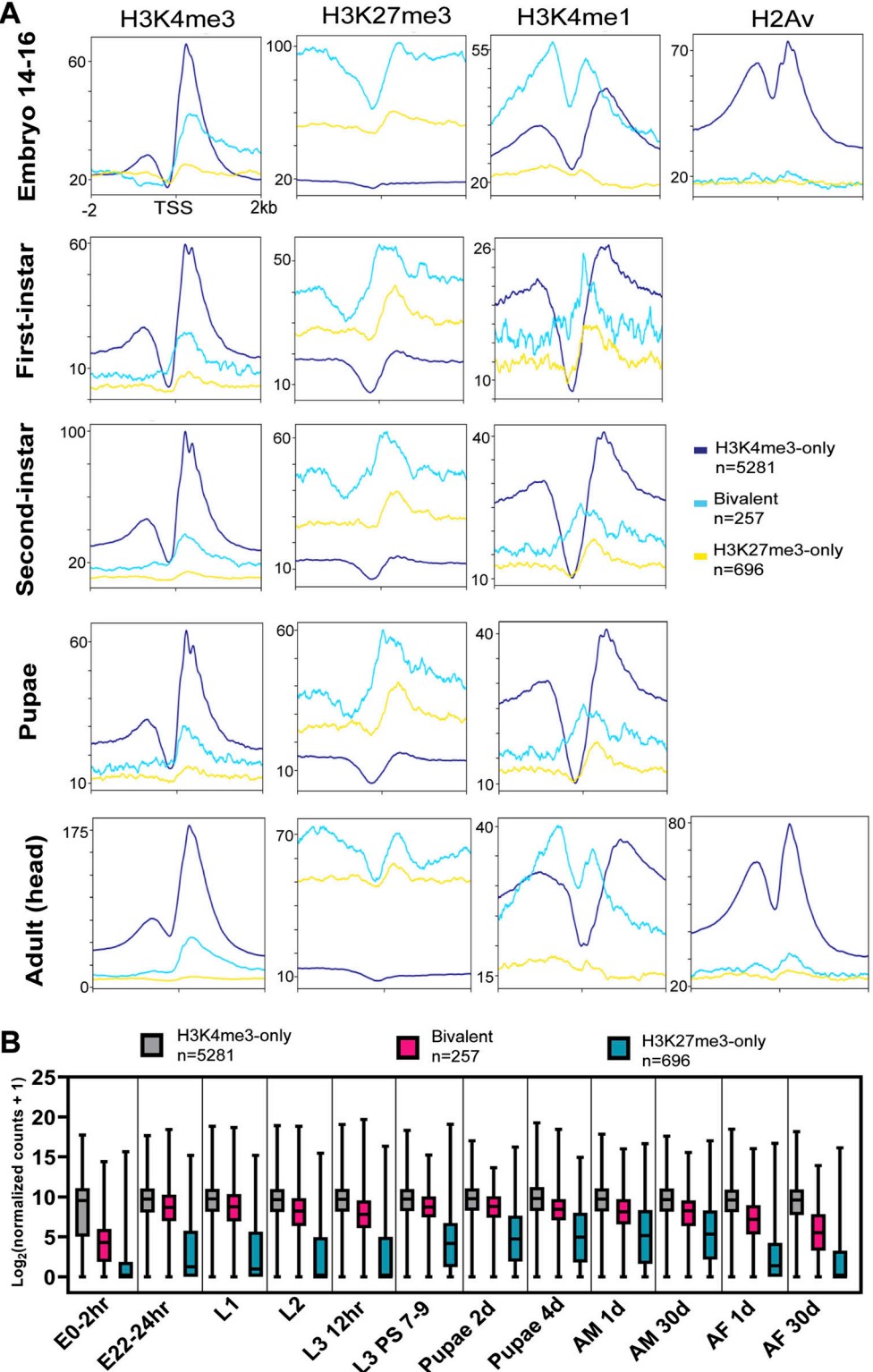

**Figure 4. Bivalency is maintained across all stages of Drosophila development.**
**(A)** Metagene plots showing the enrichment of (A) H3K4me3, H3K27me3, H3K4me1, and H2Av ChIP-seq signals active, bivalent, and silent gene promoters identified in WT third-instar larvae, during various *Drosophila* developmental stages. ChIP-seq signals are shown for regions ±2 kb from the TSS. **(B)** Box plot showing the expression levels of active, bivalent, and silent genes identified in third-instar larvae at different *Drosophila* developmental stages. Box plot: dashed center line, median; box-plot limits, upper and lower quartiles; whiskers, minimum and maximum values.

known repressive functions (Herz et al, 2012). Furthermore, Adh transcription factor 1, pipsqueak (psq), and cg, which are known for their association with PRE/TREs and regulation of polycomb binding at PREs (Huang et al, 2002; Orsi et al, 2014; Ray et al, 2016), were also highly enriched (Fig 5I). Notably, PRC1 proteins almost exclusively occupied bivalent promoters in S2 cells (Fig 5G).

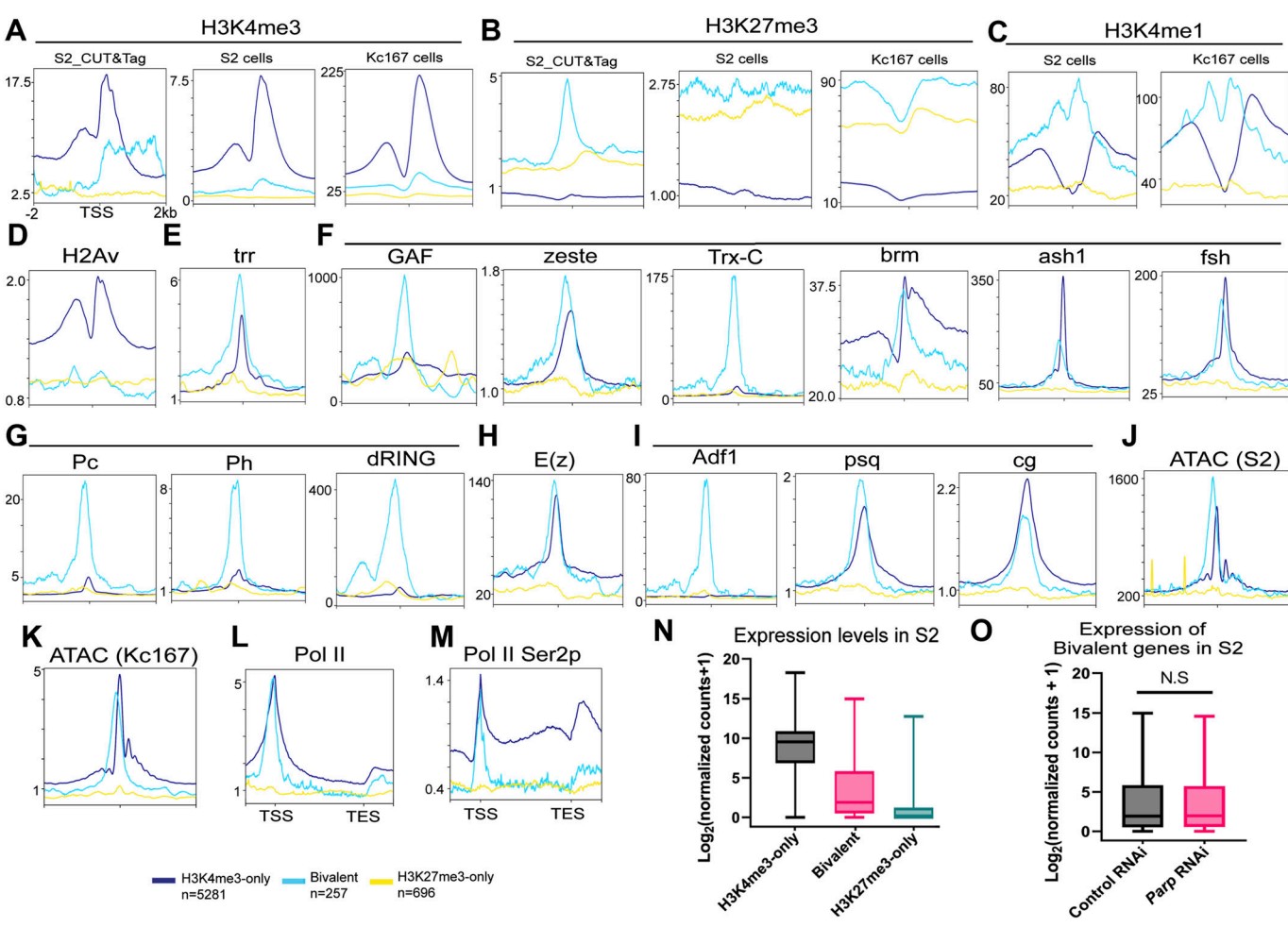

**Figure 5. Characterization of bivalency in *Drosophila* cells.**
Metagene plots showing enrichment of histone modifications, chromatin regulators, and transcription factors in S2 or Kc167 cells at the promoters of active, bivalent, and silent genes identified in third-instar larvae. **(A, B, C, D, E, F, G, H, I, J, K, L, M)** Enrichment patterns for (A) H3K4me3, (B) H3K27me3, (C) H3K4me1, (D) H2Av, trithorax group proteins: (E) trr, (F) GAF, zeste, Trx-C (C-terminal), brm, ash1, fsh, polycomb repressive complex 1: (G) Pc, Ph, dRING, polycomb repressive complex 2: (H) E(z), PRE/TRE binders, and polycomb regulators: (I) Adf1, psq, cg, (J) ATAC signals in S2 cells, (K) ATAC signals in Kc167 cells, (L) Pol II, and (M) Pol II Ser2p in S2R+. CUT&Tag, ChIP-seq, and ATAC-seq signals are shown for regions ±2 kb from the TSS. **(N)** Pol II CUT&Tag and Pol II Ser2p ChIP-seq signals are shown for the regions extending from −1 kb of the TSS to +1 kb of the TES (N) boxplot showing the expression levels of active, bivalent, and silent genes in *Drosophila* S2 cells treated with control RNAi. **(O)** Box plot showing the expression levels of bivalent genes identified in third-instar larvae in S2 cells treated with control RNAi or *Parp* RNAi (Two biological replicates). Box plot: dashed center line, median; box plot limits, upper and lower quartiles; whiskers, minimum and maximum values.

Similar to third-instar larvae, bivalent genes exhibited high accessibility in S2 cells and Kc167 cells (Fig 5J and K), despite having the highest enrichment of H3K27me3 and high PcG occupancy. The accessibility of these bivalent gene promoters may be preserved by trithorax group (TrxG) proteins and their associated chromatin remodeling activities, which would facilitate the recruitment of transcription factors and other regulatory proteins involved in TrxG/PcG-mediated gene regulation to bivalent promoters.

RNA polymerase II (Pol II) was highly occupied at bivalent promoters with levels comparable with those at H3K4me3-only promoters (Fig 5L). However, the actively elongating form of Pol II, marked by Ser2 phosphorylation (Ser2p), was significantly enriched within the gene bodies of active genes but displayed a "paused" state at the promoters of bivalent genes and was depleted at silent promoters (Fig 5M). The expression levels of bivalent genes

were substantially lower than those of silent genes but higher than those of active genes, which remained silent (Fig 5N). This observation suggests a "poised" state for bivalent genes, consistent with our previous findings in *Drosophila* at various developmental stages. Fig S5 shows CUT&Tag and ChIP-seq tracks for Phosphoribosylformylglycinamidine synthase ade2 (active), wingless wg (bivalent), and protein kinase, cAMP-dependent, catalytic subunit 2 Pka-C2 (silent) genes in S2 cells and Kc167 cells.

Next, we examined the expression of these bivalent genes using public siRNA *Parp* knock-down RNA-seq data in S2 cells (Matveeva et al, 2016) (Table S7). The expression levels of bivalent genes were not significantly different in control RNAi compared with *Parp* RNAi S2 cells (Fig 5O). Thus, although these bivalent genes have a similar profile in S2 cells and third-instar larvae, their expression may only be controlled by PARP-1 during the third-instar larvae stage.

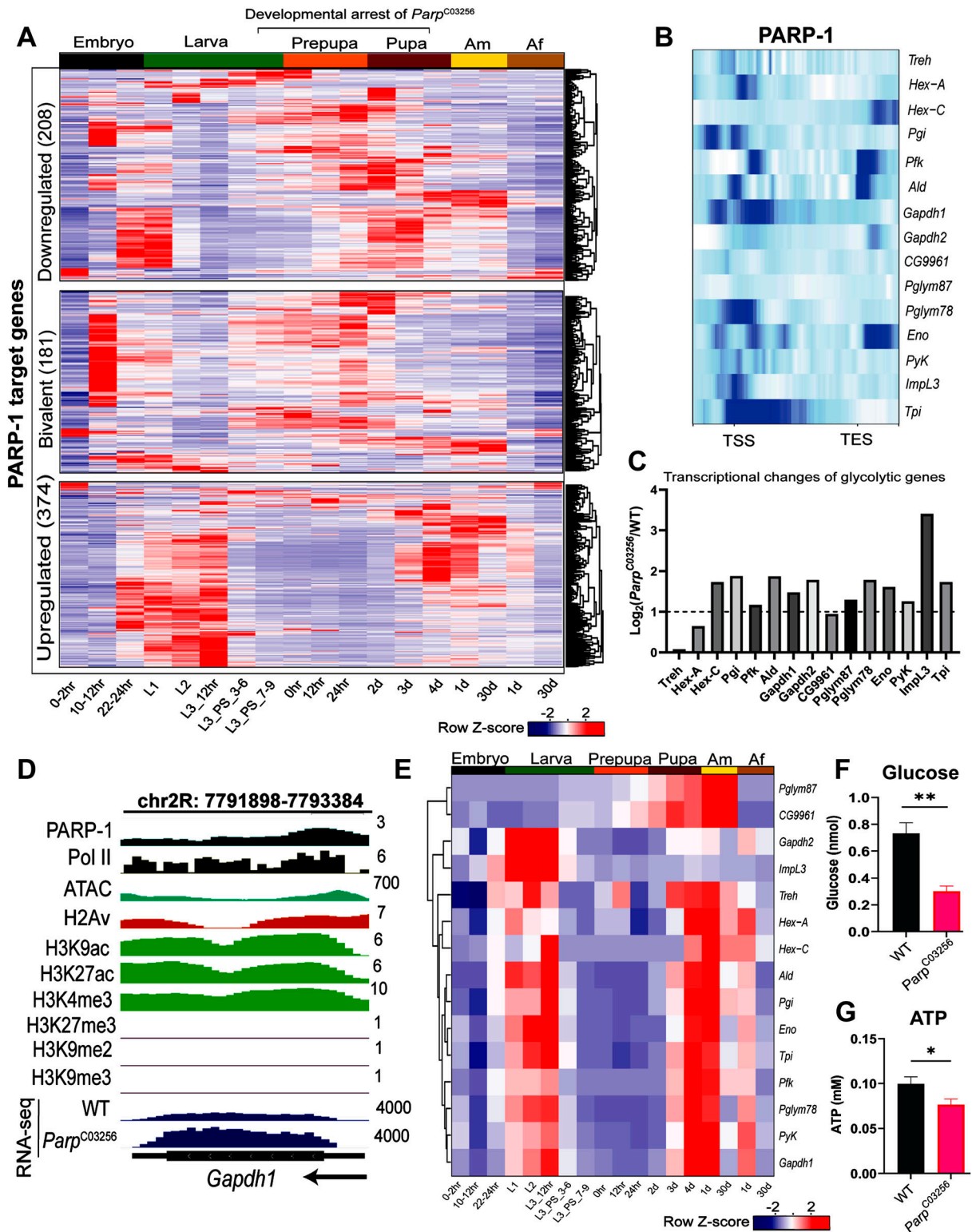

**Figure 6. PARP-1 balances metabolic and developmental gene expression during the larval-to-pupal transition.**
**(A)** Heatmap showing a temporal profile of the expression of occupied PARP-1-regulated genes (bivalent, and genes that were up-regulated or down-regulated genes in *Parp*[C03256] third-instar larvae) in WT animals during Drosophila development. Expression levels are represented as row z-scores based on normalized read counts.
**(B)** Heatmap showing PARP-1 binding at the promoters of glycolytic genes. PARP-1 ChIP-seq signals in third-instar larvae at the regions extending from −1 kb of the TSS to +1 kb TES are shown. **(C)** Expression changes of glycolytic genes (*Parp*[C03256] versus WT) in third-instar larvae. Log₂ fold changes from DESeq2 analysis are shown. Dashed line indicates a Log₂ fold change of 1 (twofold change). **(D)** IGV tracks showing normalized PARP-1, H2Av, H3K9ac, H3K27ac, H3K4me3, H3K27me3, H3K9me2, H3K9me3 ChIP-

## PARP-1 fine-tunes the expression of metabolic and developmental genes during larval-to-pupal transition

We next examined whether the expression of PARP-1-targeted genes that were differentially expressed in *Parp*^C03256^ third-instar larvae were developmentally controlled (Table S8). PARP-1 target bivalent genes and developmental genes that were down-regulated in *Parp*^C03256^ third-instar larvae were mainly expressed at the end of the embryo stage, first instar, and during the late third instar to late pupal stages, with a marked reduction of expression during the adult stages (Fig 6A). Therefore, our findings suggest that PARP-1 acts as an activator of developmental genes, including bivalent genes essential for the transition from larval to pupal stage, thus explaining the lethality observed in *Parp*^C03256^ animals during these stages.

In contrast to PARP-1-targeted developmental genes down-regulated in *Parp*^C03256^ third-instar larvae, PARP-1-targeted metabolic genes, which were up-regulated in *Parp*^C03256^ third-instar larvae, were predominantly expressed at the end of the embryo stage and during early larval stages when the animals were actively feeding. However, their expression was notably reduced between late third instar to early pupal stages; it increased again at the end of pupal stage and in adults (Fig 6A; bottom). This is consistent with previous studies that showed the expression of metabolic genes, including glycolytic genes, decreasing as the animals prepare for a non-foraging and immobile pupal life. This decrease in metabolic gene expression is an essential adaptation, as it prepares the developing animals for the unique physiological and energetic demands associated with their pupal life (White et al, 1999; Arbeitman et al, 2002). Furthermore, holometabolous insects, including *Drosophila*, have a reduced metabolic rate during metamorphosis, and it remains low until the adult flies are about to emerge (Merkey et al, 2011; Nishimura, 2020). Notably, PARP-1 binds to the promoters of glycolytic genes and represses them (Figs 6B–D and S6). As expected, glycolytic gene expression levels generally decline during larval-to-pupal transition (Fig 6E). Thus, PARP-1 tempers the expression of highly active metabolic genes in response to decreased energy demands at the onset of pupal life.

To investigate the impact of PARP-1 depletion on metabolism, we assessed glucose and ATP levels in third-instar larvae. Our results revealed a significant reduction in both glucose and ATP levels in *Parp*^C03256^ third-instar larvae (Fig 6F and G). These findings suggest that PARP-1 plays a crucial role in repressing metabolic and glycolytic genes during the transition to pupa, and this repression is essential for maintaining the normal levels of glucose and ATP required for proper metabolic homeostasis during the larval-to–pupal transition.

## Discussion

In this study, we investigated the role of PARP-1 in modulating gene expression during the third-instar larvae stage of *Drosophila*. To this end, we integrated PARP-1 ChIP-seq data in third-instar larvae and RNA-seq data of *Parp*^C03256^ mutant third-instar larvae. We identified two distinct gene expression programs regulated by PARP-1: repression of highly active metabolic genes and activation of developmental genes, including a subset of low-expression "bivalent" genes (Figs 2B–D and 3C). Notably, our results demonstrate that metabolic genes repressed by PARP-1 are normally down-regulated at the third-instar larvae stage during development, whereas the developmental genes activated by PARP-1 are active at this stage (Fig 6A). Hence, we showed that PARP-1 functions as a rheostat controlling the activation and repression of specific genes in response to developmental cues.

### Bivalency as a regulatory mechanism in *Drosophila* development and the potential role of PARP-1

The concept of bivalency has long been understood as a chromatin state that enables the regulation of developmental genes in pluripotent cells (Bernstein et al, 2006; Voigt et al, 2013; Piunti & Shilatifard, 2016). Bivalent genes, displaying both active and repressive marks, are considered to be in a poised state. This allows them to be quickly activated in response to appropriate developmental cues or environmental stimuli (Bernstein et al, 2006). This state is supposedly resolved into either H3K4me3-only for activation or H3K27me3-only for silencing upon differentiation (Bernstein et al, 2006). Interestingly, despite the traditional association of bivalency with pluripotency, recent evidence has shown that even cells with limited differentiation potential, such as mouse embryonic fibroblasts cells, exhibit a significant proportion of bivalent genes (Mikkelsen et al, 2007; Christophersen & Helin, 2010; Voigt et al, 2013). Our study raises questions about the idea that bivalency is restricted to pluripotent cells and suggests that it may persist throughout *Drosophila* development, starting from embryos, larvae, pupae, and continuing to adult flies that have less lineage specification (Fig 4A). Moreover, we observed a bivalent profile in *Drosophila* cell lines, such as S2 and Kc167 cells. The persistence of bivalency across development and cell types suggests that the regulatory mechanisms associated with bivalent chromatin marks might have a more widespread role in gene regulation than previously assumed. We postulate that cells may use bivalency to express genes at appropriate levels and modulate gene expression in response to developmental, cellular or environmental cues. In this scenario, H3K4me3 and H3K27me3 are predominantly retained at bivalent genes in *Drosophila* throughout development. However, it is important to note that our observations do not preclude the possibility that some of these bivalent genes may resolve to H3K27me3-only or H3K4me3-only states under specific conditions in certain cell types, or at different time points during development.

We hypothesize that key regulatory proteins, such as trxG/PcG complexes, and chromatin remodelers like PARP-1 play a crucial

---

seq signals, ATAC-seq signals in third-instar larvae, and RNA-seq signals in WT and *Parp*^C03256^ third-instar larvae at the indicated gene, *Gapdh1*. Black arrow indicates the direction of transcription. **(E)** Heatmap showing a temporal profile of the expression of glycolytic genes in WT animals during *Drosophila* development. Expression levels are represented as row z-scores based on normalized read counts. **(F, G)** Quantification of glucose (three biological replicates) and (G) ATP levels (five biological replicates) in WT and P*arp*^C03256^ in third instar larvae. **P < 0.01, *P < 0.05 (unpaired *t* test; two-tailed).

role in fine-tuning the expression levels of these bivalent genes in response to specific cellular signals. By maintaining a balance between activation and repression, these regulatory factors may enable dynamic shifts in gene expression in a manner essential for proper development. Consequently, bivalency may function as an important regulatory mechanism to maintain gene expression plasticity and prevent genes from irreversible silencing.

Previous studies in mammalian cells and *C. elegans* have shown that PARP-1 can recruit polycomb members to sites of DNA damage (Chou et al, 2010), raising the question of whether PARP-1 also plays a similar role at bivalent promoters. In *Drosophila* third-instar larvae, PARP-1 binds to PRE/TRE motifs, particularly the GAF motif at bivalent promoters (Fig 3E). Unlike in mammals where TrxG and PcG proteins nucleate at bivalent promoters at unmethylated CpG islands (Mikkelsen et al, 2007), *Drosophila* lacks a methylated genome, and unmethylated genomic regions are undetectable (Raddatz et al, 2013). This difference suggests that transcriptional regulators like PARP-1 might bind to PRE/TRE motifs at bivalent promoters as part of the developmental regulation in *Drosophila*.

Given this context, it remains unclear how PARP-1 is recruited to these PRE/TRE motifs in *Drosophila* and whether it can recruit TrxG/PcG proteins to bivalent promoters as well. We speculate that in response to developmental cues, PARP-1 may form a complex with GAF at bivalent promoters, thereby facilitating gene activation. However, further investigation is necessary to clarify the role of PARP-1 in the recruitment of TrxG and PcG proteins at bivalent promoters.

We showed that PARP-1 activates bivalent genes in third-instar larvae but not S2 cells (Figs 3C and 5O), which were derived from late-stage 20–24-h embryos (Schneider, 1972). As discussed earlier, the ablation of maternal stores of *Parp* RNA via RNAi in early-stage *Drosophila* embryos did not prevent them from hatching into first-instar larvae; however, they could not transition to the next stage (Tulin et al, 2002). Thus, PARP-1 may not be required for embryo development, which would explain why the expression of bivalent genes was unaffected in S2 cells treated with *Parp* RNAi. In contrast, *Parp*^C03256^ mutant animals undergo developmental arrest during the larval-to-pupal transition, a stage at which bivalent genes are highly active during development (Fig 6A; middle). Our results suggest that PARP-1 is critical for the activation of bivalent genes during this developmental transition for normal development and metamorphosis.

A possible regulatory mechanism for bivalent genes could involve the control of Pol II pausing. Pol II pausing, or stalling, typically refers to a transcriptional pause that occurs after transcription initiation and before elongation (Chen et al, 2018a; Core & Adelman, 2019). Paused Pol II is associated with the phosphorylation of serine-5 (Ser5p) at its C-terminus, a modification previously observed at bivalent promoters (Levine, 2011; Brookes et al, 2012; Ferrai et al, 2017). Our findings suggest that actively elongating Pol II may also be stalled at bivalent promoters (Fig 5N), corroborating previous studies that showed Pol II stalling at the promoter of developmental control genes using a combination of antibodies that targeted the initiating and elongating forms of Pol II (Zeitlinger et al, 2007). In another study, the elongating form of Pol II was also stalled at developmental genes (Muse et al, 2007). However, contrary reports suggest that Poll II pausing or the control of transcription

elongation does not serve as a regulatory mechanism for developmental control genes or bivalent genes (Williams et al, 2015). The apparent lack of consensus indicates that further studies will be needed to comprehensively elucidate the precise function of Poll II pausing at bivalent genes. Intriguingly, it has been demonstrated that PARP-1 facilitates the elongation of Pol II at a subset of genes in mammalian cells via ADP-ribosylation and subsequent inhibition of NELF, a protein complex that facilitates promoter-proximal pausing of Pol II (Gibson et al, 2016). Thus, PARP-1 may regulate bivalent genes by controlling Pol II elongation.

A limitation of this study concerning bivalency must be recognized. Although previous research has demonstrated a substantial overlap between "bona fide" bivalent genes identified through single-antibody ChIP-seq and sequential ChIP-seq of H3K4me3 and H3K27me3, more definitive techniques such as sequential ChIP-seq or single-cell sequencing may be required to conclusively confirm the presence of bivalent genes in *Drosophila* (Dantzer et al, 2006; Macrae et al, 2022).

### The curious case of H3K4me1 at poised promoters

H3K4me1 is predominantly recognized as an enhancer marker (Heintzman et al, 2009; Creyghton et al, 2010; Rada-Iglesias, 2018); however, recent research indicates that it is also highly enriched at poised promoters (Bae & Lesch, 2020; Yu et al, 2023). Several studies have reported two distinct enrichment patterns of H3K4me1 at promoters, whereas others have associated H3K4me1 with repression at promoters through its association with H3K27me3 (Cheng et al, 2014; Dozmorov, 2015; Bae & Lesch, 2020). Notably, unimodal H3K4me1 enrichment has been observed at bivalent promoters in pluripotent and differentiated mice and human cells (Bae & Lesch, 2020; Yu et al, 2023).

Our findings revealed that this unimodal enrichment of H3K4me1 at bivalent promoters is conserved in *C. elegans*, zebrafish sperm, *Drosophila* during development, and in S2 and Kc167 cells (Figs 4A and 5C). Nevertheless, the function of H3K4me1 at bivalent or poised promoters remains unclear. A recent study identified a H3K27me3–H3K4me1 transition at bivalent CpG island promoters, where H3K4me1 acts as a conditional repressor of tissue-specific genes during development (Yu et al, 2023). However, it is uncertain whether H3K4me1 directly influences transcription.

An intriguing hypothesis suggests that H3K4me1 enrichment at poised promoters may serve a regulatory role similar to the H3K4me1 and H2A.Z enrichment on "placeholder nucleosomes," which prevent DNA methylation encroachment during zygotic genome activation in zebrafish sperm (Murphy et al, 2018; Bae & Lesch, 2020). In this case, H3K4me1 would help maintain an epigenetically neutral state before gene activation by developmental cues (Bae & Lesch, 2020). Thus, the unimodal H3K4me1 enrichment observed at the TSS of bivalent promoters may function in a similar manner in *Drosophila*. However, as previously mentioned, there is still no evidence of DNA methylation in *Drosophila*.

We speculate that the high unimodal H3K4me1 enrichment at bivalent promoters in *Drosophila* may help demarcate poised promoters. However, several lines of evidence suggest H3K4me1

may not be instructive for transcription in both mice and *Drosophila* (Dorighi et al, 2017; Rickels et al, 2017). In fact, *Drosophila* embryos expressing a catalytically dead Trr (H3K4 monomethylase) were still able to develop to adults (Rickels et al, 2017). Thus, further studies are needed to determine the precise function of H3K4me1.

A potential complication in the "placeholder" hypothesis is the enrichment of H3K4me3 at the TSS of bivalent genes. H3K4me3 and H3K4me1 cannot coexist at the TSS in a nucleosome unless there is asymmetric enrichment, where H3K4me1 is enriched on one H3 tail and H3K4me3 on the other H3 tail of the same nucleosome. Consequently, we cannot rule out the possibility that the observed high unimodal H3K4me1 enrichment at the TSS of bivalent genes may result from cellular heterogeneity, where two populations exist: bivalent regions (H3K4me3 and H3K27me3) and silent regions (H3K4me1 and H3K27me3).

### PARP-1 as a transcriptional rheostat of developmentally regulated genes

Animals must coordinate developmental progression with metabolism appropriate for every life stage. The coordination of developmental progression with metabolism allows animals to optimize energy utilization, maintain homeostasis, and adapt to environmental changes (Koyama et al, 2020). Our results show that PARP-1 binds the promoters of metabolic genes, including glycolytic genes, and tempers their expression in preparation for pupal life (Fig 6A–E). Therefore, PARP-1 represses these genes owing to lower energy requirements at this stage. Furthermore, glucose and ATP levels were significantly reduced in *Parp*$^{C03256}$ third-instar larvae. This may be a consequence of the dysregulation of metabolic gene expression in the absence of functional PARP-1. The up-regulation of glycolytic genes in *Parp*$^{C03256}$ may lead to an imbalance in cellular metabolism, potentially causing inefficient energy production. Consequently, the reduced glucose and ATP levels in *Parp*$^{C03256}$ larvae could reflect the inability of these organisms to appropriately adjust their metabolic pathways in response to the changing energy demands associated with developmental progression.

In addition, our results indicate that low-expression developmental genes bound by PARP-1 were down-regulated in *Parp*$^{C03256}$ third-instar larvae (Fig 2D), suggesting that the loss of PARP-1–mediated regulation impacted the proper expression of genes involved in development. It is possible that the down-regulation of developmental genes in *Parp*$^{C03256}$ could also be attributed to the shift in energy allocation and resources towards the up-regulated highly active metabolic and glycolytic genes, limiting the availability of essential factors and metabolites for the expression of developmental genes. The up-regulation of highly active metabolic genes in PARP-1 mutant animals may necessitate enhanced activity of transcriptional and translational machinery, and ATP-dependent chromatin remodelers, all of which consume ATP. We postulate that this increased ATP demand may result in the use of stored energy sources, such as glycogen and lipids, followed by a surge in energy production to maintain ATP levels. However, the consequent increase in energy production may be insufficient to fully compensate for the ATP

expended during transcription, leading to a net decrease in ATP levels. This depletion in ATP levels could adversely affect the transcription of low-expression genes, such as bivalent genes.

In a related study involving mice, it was demonstrated that PARP-1 mutant mice exhibited increased energy expenditure, enhanced oxidative metabolism, and a significant up-regulation of metabolic genes (Bai et al, 2011). The researchers attributed this metabolic dysregulation to an increase in NAD$^+$ availability owing to PARP-1 ablation, which correlated with an increase in the activity of SIRT1, a histone deacetylase, and key regulator of energy metabolism that uses NAD as a substrate (Li, 2013; Chang & Guarente, 2014). Consequently, we cannot exclude the possibility that increased NAD levels in PARP-1 mutants may contribute to the metabolic dysregulation observed in these animals.

A growing body of evidence suggests that metabolic enzymes and metabolites can alter the epigenome and gene expression during development (Li et al, 2018; Miyazawa & Aulehla, 2018; Cable et al, 2021). PARP-1 may be a key factor in regulating the reciprocal relationship between metabolism and gene regulation during development. Given that PARP-1 is a ubiquitous and abundant transcriptional regulator implicated in a plethora of biological processes, our results may have far-reaching implications across many developmental and disease contexts. Collectively, then, our results establish PARP-1 as a transcriptional rheostat of developmentally regulated genes.

In summary, PARP-1 acts as a transcriptional rheostat, ensuring that specific genes respond to developmental cues. Its broad regulatory role in modulating gene expression may provide valuable insights into the complex relationship between cellular metabolism, gene regulation, and disease processes.

# Materials and Methods

### Fly husbandry and genetics

Flies were cultured on standard cornmeal–molasses agar in a 25°C incubator. All fly stocks were obtained from Bloomington Drosophila Stock Center and the Exelixis collection at the Harvard Medical School unless otherwise stated. Wandering third-instar larvae (non-foraging) were used for all experiments. *Parp*$^{C03256}$ was generated in a single pBac-element mutagenesis screen (Artavanis-Tsakonas, 2004). The UAS::PARP-1-EYFP strain was previously described (Thomas et al, 2019) and was expressed using 69B-GAL4 driver (Manseau et al, 1997). W$^{1118}$ strains that do not express YFP were used as controls for ChIP-seq and RNA-seq and termed WT/control. We used tubby balancer (TM6B) to isolate *Parp*$^{C03256}$ homozygous mutants.

### Chromatin immunoprecipitation sequencing (ChIP-seq)

75 wandering third-instar larvae (three biological replicates per genotype) were collected in a 2-ml DNA LoBind Eppendorf tube and washed twice with 1 ml 1X PBS. The larvae were homogenized in an ice-cold lysis buffer (200 µl 1X protease inhibitor cocktail,

250 μl PMSF, 800 μl 1X PBS, and 1 μl Tween 20) using a pellet pestle. The homogenized lysate was supplemented with 244.5 μl of 11% formaldehyde to a final concentration of 1.8%, and the samples were crosslinked for 15 min at room temperature on a rotator. Glycine was added to a final concentration of 500 mM to quench the fixative on ice for 5 min at room temperature. The larval debris was pelleted at 1,000g for 3 min, and the supernatant was removed. The pellet was resuspended in 1 ml sonication buffer (0.5% SDS, 20 mM Tris pH 8.0, 2 mM EDTA, 0.5 mM EGTA, 0.5 mM PMSF, and 1X protease inhibitor cocktail), and chromatin was fragmented to 300–500 bp in a Bioruptor sonicator (UCD-200) for 20 cycles (30 s of high frequency sonication, 1.5 s pause) in a cold room. The sonicated material was pelleted at 10,000g for 10 min at 4°C, supernatant was collected, and then fragment size was checked before immunoprecipitation. The sonicated chromatin was precleared and incubated with anti-GFP antibody overnight (TP-401; Torrey Pines Biolabs) at 4°C overnight. The immunoprecipitated chromatin was then collected with prewashed Protein A agarose beads for 2 h. The beads were sequentially washed with the following buffers: 1 low-salt buffer wash (0.1% SDS, 1% Triton X-100, 2 mM EDTA, 20 mM Tris–HCL pH 8.0, and 150 mM NaCl), three high-salt buffer washes (0.1% SDS, 1% Triton X-100, 2 mM EDTA, 20 mM Tris–HCL pH 8.0, 500 mM NaCl), 1 LiCL wash (2 mM EDTA, 20 mM Tris–HCl pH 8.0, 0.25 M LiCl, and 1% NP-40) and 2 TE buffer washes before elution. Bound chromatin on beads was eluted twice at room temperature using 250 μl of freshly prepared ChIP elution buffer (1% SDS, 100 mM NaHCO3) for 15 min and reverse-crosslinked overnight. The eluates were then treated with RNase A and proteinase K before DNA extraction via phenol–chloroform extraction and ethanol precipitation. Libraries were made and sequenced at Novogene.

### ChIP-seq analysis

The quality of FASTQ files (raw reads) was checked using FastQC (version. 0.11.9), and adapters were removed with fastp (Chen et al, 2018b). Trimmed FASTQ files were aligned to the *Drosophila* genome (dm6) using Bowtie2 to generate bam files (Langmead & Salzberg, 2012). Unmapped and low-quality reads were discarded from bam files (≤20 mapQuality) using BamTools (Barnett et al, 2011). Duplicate reads were identified and removed from mapped reads using Picard MarkDuplicates (http://broadinstitute.github.io/picard/). Deeptools MultiBamSummary was used to determine reproducibility of ChIP-seq reads. MACS2 was used to call peaks against control using default settings, with narrowPeaks were called for PARP-1 (third-instar larvae; L3), and PARP-1 peaks were annotated to genomic features with ChIPseeker (Yu et al, 2015). Pairwise correlation of peaks was determined using Intervene (Khan & Mathelier, 2017). MACS2 bedGraph pileups were used to generate normalized coverage of ChIP-seq signals using Deeptools bigWig-Compare by computing the ratio of the signals (IP versus Control) using a 50 bp bin size. Deeptools plotHeatmap was used to create enrichment profiles across promoters (±2 kb) in reference mode (TSS) using a 50-bp bin size. Gene enrichment profiles were determined using scaled region mode (from –1 kb upstream of the TSS to +1 kb downstream of the TES).

### Analysis of public ChIP-seq and CUT&Tag

The description of all public ChIP-seq datasets used in this study is available in the supplemental data. ChIP-seq data were analyzed as above, and narrowPeaks were called for Pol II (L3), H3K4me3 (L3), H3K9ac (L3), and H3K27ac (L3), whereas broadPeaks were called for H3K4me1 (L3), H3K9me2 (L3), H3K9me3 (L3), H3K27me3 (L3), and H2Av (L3). For GAF (salivary glands and imaginal discs of L3), Jumonji AT rich interactive domain 2 (L3), Su(z)12 (L3), and E(z) (Imaginal discs and brains of L3) ChIP-seq data, FASTQ files were quality-checked, mapped, and deduplicated as above. Deeptools bamCompare was used to generate bigwig files by computing the ratio of the signals (IP versus Control) from dereplicated bam files using a 50-bp bin size. For zebrafish sperm ChIP-seq and *C. elegans* embryo ChIP-seq, FASTQ files were quality-checked and trimmed as above and mapped to the danrer11 and ce11 genomes, respectively. H3K4me3 and H3K27me3 peaks were called as above for zebrafish sperm ChIP-seq data except the - - nomodel parameter was used to call peaks for *C. elegans* ChIP-seq data in the absence of input controls. Deeptools bamCoverage was used to generate bigwig files by normalizing to reads per kilobase per million (RPKM).

For all other ChIP-seq data, FASTQ files were quality-checked, aligned, and deduplicated, as described above. To create bigwig files, Deeptools bamCoverage was employed using default settings and normalizing to reads RPKM. Alternatively, bedgraph/bigwig files were downloaded when available. CrossMap (Zhao et al, 2014) was then used to convert bedgraph/bigwig files from dm3 to dm6 genome assembly when applicable. Deeptools plotHeatmap was used to create enrichment profiles across promoters and genes as described above.

### RNA-seq

RNA was isolated from 10 wandering third-instar larvae of WT and Parp^C03256 genetic background (three biological replicates per genotype) using RNeasy lipid tissue mini kit (QIAGEN). RNA samples were flash-frozen in liquid nitrogen and sent to Novogene for library preparation and sequencing. mRNA was purified from total RNA via poly-T oligo beads. Libraries were prepared using the Ultra II RNA library kit (NEB), and samples were sequenced on the NovaSeq 6000 platform (Illumina) at Novogene.

### RNA-seq analysis

Paired-end reads were quality-checked using FastQC and trimmed using fastp. Trimmed reads were mapped to the Drosophila genome (dm6) using RNA STAR (Dobin et al, 2013). Reads per annotated gene were counted using featureCounts (Liao et al, 2014). Differential expression analysis was performed with DESeq2 (Love et al, 2014) with $Log_2$ fold change of at least 1 (absolute) considered significant (FDR < 0.05). PARP-1 knockdown RNA-seq data in S2 cells (Matveeva et al, 2016) were analyzed as above. To visualize RNA-seq data, bam files were converted to bigwig with Deeptools bamCoverage using default parameters and normalizing to reads RPKM. For analysis of developmental time-course

gene expression, FASTQ files were trimmed and mapped as above. DESeq2 was used to generate normalized counts. Heatmap2 was used to generate heatmaps, and bivalent genes that were not expressed at any developmental stage were removed from the heatmap analysis.

### ATAC-seq analysis

Third-instar larvae ATAC-seq (Meers et al, 2018) was analyzed as follows: reads were quality-checked and groomed using fastp. Reads were then mapped to the *Drosophila* genome (dm6) with Bowtie using default parameters with analysis mode set to–very sensitive. Unmapped and low-quality reads were discarded from bam files (<=30 mapQuality), and mitochondrial reads were removed using BamTools. Duplicate mapped reads were removed with Picard MarkDuplicates. Peaks were called using MACS2 (FDR < 0.05) with default setting, except–nomodel (Build Model) and–shift size were set to –100. MACS2 bedGraph pileup was converted to bigwig for visualization using BedGraphToBigWig.

### Motif analysis

The HOMER suite was used for motif analysis (Heinz et al, 2010). Transcription factor motif analysis of PARP-1 peaks (±500 bb) was done using the findMotifsGenome.pl function with the parameters -mask -size given -S 10. De novo motif analysis of the promoters of H3K4me3-only, bivalent and H3K27me3-only genes was done using the findMotifs.pl function with the parameters -mask -S 10 -len 8,10 -start –500 -end 100.

### Annotations

The Bedtools suite was used to perform genomic arithmetic. PARP-1-targeted genes were genes with PARP-1 peaks at their promoter (±500 bp TSS). H3K4me3-only promoters had H3K4me3 peaks, but not H3K27me3 peaks. Bivalent promoters had both H3K4me3 peaks and H3K27me3 peaks at their promoter. H3K27me3-only promoters had only H3K27me3 peaks at their promoter, but not H3K4me3 peaks.

### Chip-seq, ATAC-seq, and RNA-seq visualization

IGV (2.13.1) was used to visualize bigwig files of ChIP-seq, ATAC-seq, and RNA-seq.

### Gene ontology (GO) analysis

Gene ontology terms were determined using g:profiler (FDR < 0.05) (Raudvere et al, 2019).

### Glucose and ATP measurement

Glucose and ATP concentrations were determined using the Glucose Assay Kit (ab65333; Abcam) and ATP Assay Kit (ab83355; Abcam), respectively, following the manufacturer's instructions. Before measuring the colorimetric signals at 570 nm, samples were deproteinized

with a Deproteinizing Sample Preparation Kit (ab204708; Abcam). Signal quantification was carried out using a Biotek Cytation 3 Imager.

### Statistical analyses

Results were analyzed using the indicated statistical test in GraphPad Prism (9.4.0). Statistical significance of ChIP-seq and ATAC-seq peaks was determined using MACS2. Q-value cutoffs for RNA-seq and GO analysis were determined with DESeq2 and G:profiler, respectively.

## Data Availability

All data generated or analyzed during this study are included in this article (Table S2). No custom code was generated in this study. ChIP-seq and RNA-seq data generated in this study are available on the GEO database (Accession no: GSE217730 and GSE222877). https://www.ncbi.nlm.nih.gov/geo/query/acc.cgi?acc=GSE222877 PARP-1 and other third-instar larvae (ChIP-seq) can be visualized on the UCSC genome browser via: https://genome.ucsc.edu/s/Gbolahan/PARP1_L3_ChIP%2Dseq.

## Supplementary Information

## Acknowledgements

We thank Drs. Benjamin Roche and Yaroslava Karpova for their comments and critical reading of earlier versions of the article. This study was supported by the Department of Defense grants PC160049 and the National Science Foundation MCB-2231403 to A Tulin and an International Training Scholarship by the American Society for Cell Biology through the International Federation for Cell Biology to G Bamgbose.

### Author Contributions

G Bamgbose: conceptualization, resources, data curation, formal analysis, validation, investigation, methodology, and writing—original draft, review, and editing.
A Tulin: conceptualization, resources, supervision, funding acquisition, validation, investigation, methodology, project administration, and writing—review and editing.

### Conflict of Interest Statement

The authors declare that they have no conflict of interest.

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
