## [Reviewer comments · Life Science Alliance]

Life Science Alliance

PARP-1 is a transcriptional rheostat of metabolic and bivalent genes during development

Alexei Tulin and Gbolahan Bamgbose

DOI: <https://doi.org/10.26508/lsa.202302369>

Corresponding author(s): Alexei Tulin, University of North Dakota

Review Timeline:

Submission Date:	2023-09-13
Editorial Decision:	2023-11-03
Revision Received:	2023-11-14
Editorial Decision:	2023-11-16
Revision Received:	2023-11-19
Accepted:	2023-11-20

Transaction Report:

November 3, 2023

Re: Life Science Alliance manuscript #LSA-2023-02369-T

Prof. Alexei Tulin
University of North Dakota
501 North Columbia Road, Stop 90
GRAND FORKS, ND 582028367

Dear Dr. Tulin,

Thank you for submitting your manuscript entitled "PARP-1 is a transcriptional rheostat of metabolic and bivalent genes during development" to Life Science Alliance. The manuscript was assessed by expert reviewers, whose comments are appended to this letter. We invite you to submit a revised manuscript addressing the Reviewer comments.

Thank you for this interesting contribution to Life Science Alliance. We are looking forward to receiving your revised manuscript.

Sincerely,

B. MANUSCRIPT ORGANIZATION AND FORMATTING:

Reviewer #1 (Comments to the Authors (Required)):

This manuscript by Tulin and colleague is impressive. This manuscript describes the critical role of PARP in *Drosophila* differentiation. The differentiation process in *Drosophila* is very different than that of mammals. Thus the authors could rearrange the manuscript to present as a function of the various developmental stages of *Drosophila*. This is an excellent manuscript that complements well studies during mammalian development.

Reviewer #2 (Comments to the Authors (Required)):

In this manuscript, Bamgbose and Tulin study the transcriptional role of PARP-1 during development in *Drosophila* using ChIPseq and RNAseq approaches in wild-type and *Parp* mutant third-instar larvae. They found that PARP-1 activate bivalent genes during development, maintaining the balance between metabolic and developmental gene expression, allowing proper progression through developmental stages. The manuscript is well writing and the results are conclusive. Some concerns should be addressed.

1.- Does the transcriptional role of PARP-1 during development depend on its enzymatic activity? If the PARP-1 selective inhibitor AZD5305 inhibits PARP-1 in *Drosophila*, it might be interesting to explore this question.

2.- How is PARP-1 recruited to the bivalent promoter?

Minor comments

1.- Figure 1: The order of figures 1D and 1E should be changed.

2.- The author states in the discussion that " recent evidence has shown that even cells with limited differentiation potential, such as mouse embryonic fibroblasts and T cells, exhibit a significant proportion of bivalent genes". However, T cells are a cell population with a high differentiation potential during the immune response. Consequently, this statement should be modified. Indeed, the presence of a significant proportion of bivalent genes in T cells would be associated with their high differentiation potential.

Dear reviewers of our manuscript,

I hope this letter finds you well. We would like to express our gratitude for the time and effort that the two reviewers have invested in evaluating our manuscript titled "PARP-1 as a transcriptional rheostat of metabolic and bivalent genes during development." Their feedback has been instrumental in refining our study, and we have addressed each comment to ensure the revised manuscript meets the standards of Life Science Alliance.

Please find below our detailed response to the reviewers' comments.

Reviewer 1

This manuscript by Tulin and colleague is impressive. This manuscript describes the critical role of PARP in *Drosophila* differentiation. The differentiation process in *Drosophila* is very different than that of mammals. Thus the authors could rearrange the manuscript to present as a function of the various developmental stages of *Drosophila*. This is an excellent manuscript that complements well studies during mammalian development.

Response: We thank the Reviewer for his/her positive comments on our manuscript. PARP-1 mutant animals used in this study undergo developmental arrest during the larval-pupal transition. Thus, our goal was to understand PARP-1's transcriptional role during this critical stage of *Drosophila* development. Also, it is during this transition that PARP1's enzymatic activity is at its highest during *Drosophila* development (1). This is why the manuscript was written to focus on PARP-1's role at this stage instead of the different stages of *Drosophila* development.

Reviewer 2

General Comments

In this manuscript, Bamgbose and Tulin study the transcriptional role of PARP-1 during development in *Drosophila* using ChIPseq and RNAseq approaches in wild-type and Parp mutant third-instar larvae. They found that PARP-1 activate bivalent genes during development, maintaining the balance between metabolic and developmental gene expression, allowing proper progression through developmental stages. The manuscript is well writing and the results are conclusive. Some concerns should be addressed.

Response: We thank the Reviewer for their positive comments on our manuscript. We have addressed their concerns below:

Major comments:

1.- Does the transcriptional role of PARP-1 during development depend on its enzymatic activity? If the PARP-1 selective inhibitor AZD5305 inhibits PARP-1 in *Drosophila*, it might be interesting to explore this question.

Response: In our previous studies, we have demonstrated that PARP1's ability to parylate chromatin-associated proteins, leading to chromatin relaxation, is necessary to facilitate transcription in *Drosophila* during development (2–5). This underscores the importance of PARP1's enzymatic activity in its transcriptional regulatory functions.

2.- How is PARP-1 recruited to the bivalent promoter?

Response: We thank the Reviewer for this interesting question. We believe this is beyond the scope of the manuscript. However, we have updated our discussion as follows:

Previous studies in mammalian cells and *C. elegans* have shown that PARP-1 can recruit Polycomb members to sites of DNA damage (6), raising the question of whether PARP-1 also plays a similar role at bivalent promoters. In *Drosophila* third-instar larvae, PARP-1 binds to PRE/TRE motifs particularly the GAF motif at bivalent promoters (Figure 3E). Unlike in mammals, where TrxG and PcG proteins nucleate at bivalent promoters at unmethylated CpG islands (7), *Drosophila* lacks a methylated genome, and unmethylated genomic regions are undetectable (8). This difference suggests that transcriptional regulators like PARP-1 might bind to PRE/TRE motifs at bivalent promoters as part of the developmental regulation in *Drosophila*.

Given this context, it remains unclear how PARP-1 is recruited to these PRE/TRE motifs in *Drosophila* and whether it can recruit TrxG/PcG proteins to bivalent promoters as well. We speculate that in response to developmental cues, PARP-1 may form a complex with GAF at bivalent promoters, thereby facilitating gene activation. However, further investigation is necessary to clarify PARP-1 in the recruitment of TrxG and PcG proteins at bivalent promoters.

Minor comments

1.- Figure 1: The order of figures 1D and 1E should be changed.

Response: We thank the Reviewer for pointing this out. We have changed the order of figures 1D and 1E.

2.- The author states in the discussion that " recent evidence has shown that even cells with limited differentiation potential, such as mouse embryonic fibroblasts and T cells, exhibit a significant proportion of bivalent genes". However, T cells are a cell population with a high differentiation potential during the immune response. Consequently, this statement should be modified. Indeed, the presence of a significant proportion of bivalent genes in T cells would be associated with their high differentiation potential.

Response: We thank the Reviewer for the correction. We have modified the statement as follows:

Interestingly, despite the traditional association of bivalency with pluripotency, recent evidence has shown that even cells with limited differentiation potential, such as mouse embryonic fibroblasts cells, exhibit a significant proportion of bivalent genes (9–11)

References

1. E. Kotova, M. Jarnik, A. V. Tulin, Poly (ADP-Ribose) Polymerase 1 Is Required for Protein Localization to Cajal Body. *PLOS Genet.* **5**, e1000387 (2009).
2. A. Tulin, D. Stewart, A. C. Spradling, The *Drosophila* heterochromatic gene encoding poly(ADP-ribose) polymerase (PARP) is required to modulate chromatin structure during development. *Genes Dev.* **16**, 2108–2119 (2002).
3. C. J. Thomas, E. Kotova, M. Andrade, J. Adolf-Bryfogle, R. Glaser, C. Regnard, A. V. Tulin, Kinase-Mediated Changes in Nucleosome Conformation Trigger Chromatin Decondensation via Poly(ADP-Ribosyl)ation. *Mol. Cell.* **53**, 831–842 (2014).

4. Y. Ji, A. V. Tulin, The roles of PARP1 in gene control and cell differentiation. *Curr. Opin. Genet. Dev.* **20** (2010), pp. 512–518.
5. A. Tulin, A. Spradling, Chromatin loosening by poly(ADP)-ribose polymerase (PARP) at *Drosophila* puff loci. *Science* (80-.). **299**, 560–562 (2003).
6. D. M. Chou, B. Adamson, N. E. Dephoure, X. Tan, A. C. Nottke, K. E. Hurov, S. P. Gygi, M. P. Colaiácovo, S. J. Elledge, A chromatin localization screen reveals poly (ADP ribose)-regulated recruitment of the repressive polycomb and NuRD complexes to sites of DNA damage. *Proc. Natl. Acad. Sci. U. S. A.* **107**, 18475–18480 (2010).
7. T. S. Mikkelsen, M. Ku, D. B. Jaffe, B. Issac, E. Lieberman, G. Giannoukos, P. Alvarez, W. Brockman, T. K. Kim, R. P. Koche, W. Lee, E. Mendenhall, A. O'Donovan, A. Presser, C. Russ, X. Xie, A. Meissner, M. Wernig, R. Jaenisch, C. Nusbaum, E. S. Lander, B. E. Bernstein, Genome-wide maps of chromatin state in pluripotent and lineage-committed cells. *Nat. 2007 4487153.* **448**, 553–560 (2007).
8. G. Raddatz, P. M. Guzzardo, N. Olova, M. R. Fantappiè, M. Rampp, M. Schaefer, W. Reik, G. J. Hannon, F. Lyko, Dnmt2-dependent methylomes lack defined DNA methylation patterns. *Proc. Natl. Acad. Sci. U. S. A.* **110**, 8627–8631 (2013).
9. N. S. Christophersen, K. Helin, Epigenetic control of embryonic stem cell fate. *J. Exp. Med.* **207**, 2287–2295 (2010).
10. T. S. Mikkelsen, M. Ku, D. B. Jaffe, B. Issac, E. Lieberman, G. Giannoukos, P. Alvarez, W. Brockman, T. K. Kim, R. P. Koche, W. Lee, E. Mendenhall, A. O'Donovan, A. Presser, C. Russ, X. Xie, A. Meissner, M. Wernig, R. Jaenisch, C. Nusbaum, E. S. Lander, B. E. Bernstein, Genome-wide maps of chromatin state in pluripotent and lineage-committed cells. *Nat. 2007 4487153.* **448**, 553–560 (2007).
11. P. Voigt, W. W. Tee, D. Reinberg, A double take on bivalent promoters. *Genes Dev.* **27**, 1318–1338 (2013).

November 16, 2023

RE: Life Science Alliance Manuscript #LSA-2023-02369-TR

Prof. Alexei Tulin
University of North Dakota
501 North Columbia Road, Stop 90
GRAND FORKS, ND 582028367

Dear Dr. Tulin,

Thank you for submitting your revised manuscript entitled "PARP-1 is a transcriptional rheostat of metabolic and bivalent genes during development". We would be happy to publish your paper in Life Science Alliance pending final revisions necessary to meet our formatting guidelines.

- please consult our manuscript preparation guidelines <https://www.life-science-alliance.org/manuscript-prep> and make sure your manuscript sections are in the correct order
- Please upload all figure files as individual ones, including the supplementary figure files
- please add the Twitter handle of your host institute/organization as well as your own or/and one of the authors in our system
- please add a callout for Figures Fig 3F, Fig 4B, Fig S1A,B, Fig S4A,B,C,D to your main manuscript text

A. FINAL FILES:

B. MANUSCRIPT ORGANIZATION AND FORMATTING:

Sincerely,

November 20, 2023

RE: Life Science Alliance Manuscript #LSA-2023-02369-TRR

Prof. Alexei Tulin
University of North Dakota
501 North Columbia Road, Stop 90
GRAND FORKS, ND 582028367

Dear Dr. Tulin,

Thank you for submitting your Research Article entitled "PARP-1 is a transcriptional rheostat of metabolic and bivalent genes during development". It is a pleasure to let you know that your manuscript is now accepted for publication in Life Science Alliance. Congratulations on this interesting work.

DISTRIBUTION OF MATERIALS:

Again, congratulations on a very nice paper. I hope you found the review process to be constructive and are pleased with how the manuscript was handled editorially. We look forward to future exciting submissions from your lab.

Sincerely,
